# Public perceptions of biospecimen sampling and uncertainty in the context of personalised nutrition

Katharine Lee[1]*, Estelle Corbett[2], Rebecca Hafner[3], Julie Barnett[4]

**1** Dept. of Psychology, University of Bath, Claverton Down, United Kingdom, **2** Dept. of Psychology, University of Bath, Claverton Down, United Kingdom, **3** School of Psychology, University of Swansea, Singleton Campus, Swansea, United Kingdom, **4** Dept. of Psychology, University of Bath, Claverton Down, United Kingdom

* kl614@bath.ac.uk

## Abstract

Personalised nutrition based on analysis of biospecimen generates individual-specific dietary recommendations and potentially, improved health. However, the science underpinning these approaches is evolving and uncertain. Additionally, users must provide a biological sample appropriate to the analytic approach being taken. This two-part quasi-experimental study sought to understand the impact of certainty and sample type on affective responses and attitudes to personalised nutrition. Participants (n716) completed a free association task and an attitudinal survey. Participants responded with more positive affect and attitudes to personalised nutrition when the science was characterised as certain. Attitudes to personalised nutrition were not affected by sample type, although contemplating providing a stool sample elicited more negative affective responses than other samples. This suggests that the need to provide a stool sample could be a barrier to microbiome-based personalised nutrition. We consider the implications of our findings in relation to future research and to providers of personalised nutrition.

## Introduction

There is increasing research, practitioner, and commercial attention to developments in personalised nutrition (PN) and their potential to drive behavioural changes that are conducive to improved health [1]. Although no single definition of PN exists, Ordovas, Ferguson [2] suggest it is a tripartite approach that: "Uses information on individual characteristics to develop targeted nutritional advice, products, or services to assist people to achieve a lasting dietary change in behaviour that is beneficial for health" (pp. 1). The acceleration of recent interest is associated with moving from lifestyle (e.g., diet, income, life-stage, or household composition) and phenotypic information (e.g., weight, cholesterol, or other indicators of health status) [3] to the increased

**Data availability statement:** Bath: University of Bath Research Data Archive. https://doi.org/10.15125/BATH-01605.

**Funding:** This project has received funding from the European Union's Horizon 2020 research and innovation programme under grant agreement No 816303 (https://europa.eu/european-union/abouteu/ symbols/flag_en). The funders had no role in study design, data collection and analysis, decision to publish, or preparation of the manuscript.

**Competing interests:** The authors have declared that no competing interests exist.

precision promised by genetic, genomic, metabolomic, or gut microbiome analyses [4]. However, the evidence for whether personalised nutrition, over general dietary recommendations, leads to superior health measurements is mixed [5,6]. The exact nature of the relationship between the science and tailored PN advice and consequent health outcomes services is highly uncertain, due to the evolving nature of the science itself, the complexities and heterogeneity of methodologies and technologies [7], as well as the uncertainty inherent in the inter-individual variability in responses to interventions [8]. Research on public attitudes and inclinations to PN has increased in recent years but thus far the extent to which these are contingent on appreciation of the associated uncertainties remains largely unexamined. The analyses that hold potential, albeit uncertain, promise for tailored dietary recommendations have shifted from being based on lifestyle and phenotype information to the collection and provision of biomarker or genetic information [4]. This raises the question of whether the nature of the sample collection process has any effect on public perceptions of, and intentions to engage with, PN products and services. Using biochemical markers can overcome some of the unavoidable inaccuracies that derive from self-reported food intake [9] but given that they are based on the provision of (e.g.,) blood, urine – or, in the case of the microbiome – stool samples, here too, we can expect the views of consumers to relate to the appetite for PN products and services.

To set the scene for an examination of the effect of uncertainty and of the nature of the required sample on perceptions of personalised nutrition, this introduction will unfold as follows. After a brief overview of what we know about public perceptions of PN, the case for considering the possible impacts of uncertainty and of the sample collection process on attitudes to PN will be considered. Finally, the research questions will be outlined.

### Public responses to personalised nutrition

Previous research has indicated that consumers generally regarded PN positively where it was associated with benefits such as weight loss, muscle gain, and disease prevention [10,11]. However, participants also associated PN with individuals at higher disease risk [10]. Motivations for engaging with PN include health management, personal appearance, and athletic performance [10] as well as curiosity and scientific advances [12].

Barriers to PN uptake include willpower, price, food preferences, limited choices, and convenience [10,13,14] as well as some scepticism about the science and scientific evidence [12]. The online delivery of PN services may be a barrier [10,11,15] where the interface is seen as hard to navigate or if it undermines the believability of the science of PN [13]. The collection of sensitive biological samples and requirements for personal information may be associated with heightened privacy and data security concerns [10,11,16] although consumers may equate donating a personal sample with improved diagnostic precision. In line with this, Nielsen and El-Sohemy [17] found individuals are more likely to follow dietary recommendations when they are personalised in line with their genetic profile.

Although public perceptions to PN are sensitive to a range of issues related to risk and benefit, the impact of uncertainty is thus far unexplored, yet, as the next section will suggest, there is every reason to suggest that the presentation of (un)certainty will be related to the perceived acceptability of PN.

### Navigating the impact of scientific uncertainty

**Uncertainty of personalised nutrition.** The scientific underpinnings of PN are unavoidably characterised by uncertainty because of continuously evolving, sometimes contradictory, knowledge, and complex and heterogeneous methodologies and technologies. Additionally, most data originate from observational studies or animal experiments [7,18]. Uncertainty is particularly pronounced in genetic-based PN due to the intricate and multifaceted interactions between an individual's genetic makeup, diet, and health outcomes [19,20] and its effectiveness may vary for different populations [7]. While associations between genotypes, microbiota, and health outcomes are increasingly recognized [21,22], the precise nature of these interactions and their implications remain only partially understood [23]. For instance, the gut microbiome (GM), which responds dynamically to various environmental factors, is one focal point of PN, yet the nature and extent of its interactions with genes and diet and their impact on health is uncertain [24]. The diversity and composition of the GM have been linked to health outcomes, but a definitive "healthy" GM is context-dependent and individual-specific [25] with some individuals 'responding' and others 'not-responding' to microbiome-based dietary interventions [26].

**The communication of scientific (un)certainty.** Given that both consumers and professionals may have concerns about scientific validity, fraudulent practices, and the accuracy of tests [16], the communication of scientific uncertainty around PN requires careful attention. Whilst some experimental research found that communicating uncertainty increases doubt and reduces perceived credibility or intention to partake, other studies found it led to increased trustworthiness and more positive attitudes [27].

Perhaps paradoxically, communicating uncertainty about scientific claims of threat can increase source trust and credibility [28]. People tend to be more accepting of uncertainty associated with the scientific risk management process than government inaction, emphasizing the importance of promptly disclosing uncertainty [29]. According to Johnson and Slovic [28], the communication of uncertainty can either enhance source credibility or signal incompetence. Qualitative research suggests that, in an environment of low trust, disclosure of uncertainty may signal a lack of accountability rather than a transparent approach to risk management [30] yet, denials of uncertainty and claims of absolute safety are more likely to be mistrusted than admissions of uncertainty [31].

In a scoping review examining uncertainty-related communication during the COVID-19 pandemic, Ratcliff, Wicke [32] noted that communicating or experiencing uncertainty often had either neutral or negative effects on trust, credibility, and support for science. However, in a different domain – precision medicine – overall, manipulating scientific and data-related uncertainty had no impact on attitudes [33], though for those with a higher level of scientific understanding or a greater tolerance for uncertainty, disclosing uncertainty had a positive effect on trust and attitudes toward participation in a precision medicine research project. The challenges of communicating uncertainty are exacerbated in the context of PN, occupying as it does a space between the 'medical' and the 'consumer'. This blending of medical and lifestyle contexts, each subject to different regulatory regimens [34] foregrounds questions about transparency, trust and the implications for consumers' perception, comprehension, and acceptance of these products and the associated procedures – such as sample collection.

**Collection of biological samples to inform personalised nutrition.** Given that increasingly PN is based on biological samples, it is also important to consider how perceptions of PN might vary in respect of the nature of the required sample and the associated methods of sample collection.

Alongside lifestyle measures, the reliance on bio-specimen sampling has become a prominent methodology in PN. The choice of sample type depends on factors such as invasiveness, data quality, and relevance to the metabolic processes under investigation [4]. Some biological samples are used to gather information about an individual's phenotype

(i.e., biochemical markers for nutritional status) such as blood, urine, saliva, cheek swab, faeces, sweat, hair and exhaled breath. Multiple types of bio-specimens can be used interchangeably to provide the same information; for certain types of personalised nutrition, specific biospecimens are needed to gather information uniquely present in that sample, for example, when PN is based on an analysis of the microbiome, a stool sample is needed to analyse the DNA of the microbes present [4].

Although rarely explored in relation to nutrition, varying public attitudes towards donating different biological samples have been observed in several studies. For instance, in the context of cancer survivor nutrition research, Keaver, McGough [35] found that individuals were more willing to provide oral swabs, urine and blood samples than stool samples. In the field of nutrition interventions, Khalsa, Burton [36] reported higher return rates for hair and urine samples compared to stool samples though the differences were not significant. Similar trends extend beyond nutrition research. Biobank studies demonstrated a preference for donating saliva, followed by blood and stool samples [37]. Surveys have highlighted the variability in willingness to donate various biological samples [35,38] though most studies reporting on perceptions toward and willingness to donate biological samples are conducted in the context of particular diseases states and/or biobank research [39–41] rather than PN for disease prevention and overall health. Therefore, in addition to exploring the impact of scientific uncertainty, we will seek to understand public attitudes toward donating biological samples in the context of PN. We will do so through two linked studies:

- In study 1, using a thought listing task and associated affect ratings we examine participants' spontaneous reactions to different biological samples in the context of either certain or uncertain science.

- In study 2 we explore whether the expression of scientific certainty and type of biological sample influences ratings of ethical concerns, attitudes about, and intentions towards, personalised nutrition.

## Materials and methods

### Design

Studies 1 and 2 were embedded in a quasi- experimental between-subjects 2 (certain v. uncertain) x 3 (urine v. blood v. stool) design. Scientific certainty and sample type was presented and manipulated within a short vignette. This was followed by a free association task (study 1) where the outcomes of interest were the words generated by participants in response to the vignette and ratings of affect participants associated with each word. For study 2, the impact of varying uncertainty and sample type on participants attitudes to PN was assessed using a series of outcome measures. A favourable ethical opinion for the study was granted by University of Bath Psychology Research Ethics Committee (ref: 21–254).

### Sample

We recruited a sample of 716 UK participants via the Prolific platform. Prolific is a well-established platform that helps researchers recruit participants for their online research. It enables researchers to recruit samples according to their research needs, in our case, we recruited a nationally representative sample. The effectiveness of the (un)certainty manipulation was checked by asking participants to indicate their agreement with the following statement "The science behind personalised nutrition is certain" on a 5-point scale (1 = completely disagree, 5 = completely agree). Only participants expressing a view about certainty aligned to the certainty condition they were allocated to were retained for analysis, i.e., those in the certainty condition who answered 4 or more ('agree' and 'completely agree') and those in the uncertainty condition who answered 2 or less ('disagree' and 'completely disagree'). 261 participants were retained for analysis: 183 participants were in the Certainty condition and 78 in the Uncertainty condition. For the sample conditions, 94 participants were in the Urine condition, 81 participants were in the Blood condition, and 86 participants were in the Stool condition. Forty eight percent were male. Demographic information is available in the supplementary information file.

## Materials and procedure

Data were collected via the Prolific platform in February 2022. Participants first read through an information form and gave their consent via a tick box. Failure to tick the box would mean that potential participants would not progress to participation in the study. Following some demographic questions, participants read a short passage about personalised nutrition based on a type of classification called metabotyping. This was chosen because it is a type of personalised nutrition that may require different biological samples. The explanation of metabotyping referred to the identification of individual characteristics that would lead to targeted nutritional advice that aimed to improve health. The manipulation was relatively subtle, but in the certainty condition, definitive language was used (e.g., diet **is** an important contributor; it **is** possible to design targeted dietary advice). In the uncertainty condition, more provisional language was used (e.g., diet *is thought to be* an important contributor; it *may be possible* to design targeted dietary advice). Immediately following the information relating to the certainty of the science, participants were directed to one of three biological sample conditions, either Urine, Blood, or Stool, to read about providing one of these samples. Here, we adapted (shortened, simplified as far as feasible) UK NHS guidelines for collecting each of the samples available at the time. We outlined the process for collecting each type of sample in a series of short steps, from washing hands, to carrying out the sampling procedure, through to packaging and posting it off for analysis.

**Study 1: Word association.** Having been randomly allocated to one of the six conditions, immediately after reading the vignette, participants were directed to the word association task. This was based on a study by Slovic, Layman [42]. Participants were invited to write the three words that came to mind when thinking about collecting the biological sample they had read about and were asked to rate each of the words they had provided to indicate the associated affect from 1 – very negative, to 5 – very positive, with 3 representing a neutral response. The words and associated affect ratings were recorded for analysis.

**Analysis.** Using a summative approach to qualitative content analysis [43], an initial broad coding of the words that participants provided was conducted to categorise the words provided based on their meaning. Over several iterations, words were subject to further lemmatisation, whereby synonyms were grouped together, and the words placed into broader categories of meaning. The research team consulted on category development throughout the process. A total of 21 unique word categories were identified. Counts were performed for each category within each of the six condition permutations. Median affect ratings for the six conditions were calculated. Statistical analyses were conducted to identify the effect of certainty and sample condition on median affect ratings.

**Study 2: survey measures.** Scales were developed to measure three ethical dimensions – data security and privacy, the veracity of PN claims and equality of access to PN. Existing validated scales were selected to measure attitudes towards PN, perception of benefits of PN, intention to adopt PN, and perceived efficacy of trust and regulation of PN [44]. Table 1 details these scales, their provenance, the items, and scale reliabilities.

**Analysis.** Where necessary, scales were reversed prior to conducting the analysis, so that higher numbers always indicated more positive responses, or reduced concern. Due to assumptions of normality not being met, non-parametric tests were conducted to identify the effects of (un)certainty and sample manipulations on each dependent variable.

## Results

### Study 1a Results of word association study

A breakdown of the number of participants and the total number of words elicited in each is outlined in Table 2. Participants produced very similar numbers of words in all conditions.

Following lemmatisation, the elicited words were coded into 21 discrete categories, described and detailed in Table 3 below.

As the assumptions of normality were not met, a Kruskal-Wallis H test was used to test for any differences between the median affective responses in the two certainty and three sample type conditions. For the sample condition, pairwise comparisons were used to locate where any significant differences lay between the affect rating of the three samples.

**Table 1. Scales used in survey study.**

| Name of scale | Question asked | Items | Response | Reliability (Cronbach's alpha) |
|---|---|---|---|---|
| *Ethical scales* | | | | |
| PN data security and privacy (Developed for this study) | If you decided to use a personalised nutrition service to give you dietary advice, how much would you be concerned about… | Whether your data would remain private<br>Whether your data would be stored securely<br>Whether correct consent procedures would be followed<br>Whether your data might be used inappropriately | Four-point scale, from extremely concerned to not at all concerned | .940 |
| Veracity about PN claims (developed for this study) | If you decided to use a personalised nutrition service to give you dietary advice, how much would you be concerned about… | Whether you could believe all the benefit claims being made about personalised nutrition<br>Whether claims about personalised nutrition can be trusted<br>Whether personalised nutrition is more effective than general nutritional advice<br>Whether personalised nutrition advice given would be genuinely beneficial | Four-point scale, from extremely concerned to not at all concerned | .908 |
| Equality of access to PN (developed for this study) | If you decided to use a personalised nutrition service to give you dietary advice, how much would you be concerned about… | Whether personalised nutrition only benefits those who can afford it<br>Whether everyone who could benefit from personalised nutrition would be able to access it | Four-point scale, from extremely concerned to not at all concerned | .857 |
| *Other scales* | | | | |
| Attitude towards personalised nutrition (adapted from Poínhos et al. (2014)) | Personalised nutrition is: | Worthless to Valuable Unpleasant to Pleasant<br>Boring to Interesting<br>Bad to Good<br>Untrustworthy to Trustworthy<br>Ineffective to Effective | Five-point scale, where 5 is most and 1 least positive | .879 |
| Benefit perception associated with personalised nutrition (adapted from Poínhos et al. (2014)) | Please indicate the extent to which you agree or disagree with the following statements | Personalised nutrition could benefit me personally<br>Personalised nutrition could benefit my family<br>Personalised nutrition could benefit an average member of the society in which I live | Five-point scale, from completely disagree to completely agree | .887 |
| Intention to adopt personalised nutrition (adapted from Poínhos et al. (2014)) | Please indicate the extent to which you agree or disagree with the following statements | I intend to adopt personalised nutrition<br>I would consider adopting personalised nutrition<br>I am definitely going to adopt personalised nutrition | Five-point scale, from completely disagree to completely agree | .893 |
| Efficacy of trust and regulation (adapted from Poínhos et al. (2014)) | Please indicate the extent to which you agree or disagree with the following statements | I worry that a personalised nutrition diet plan is not effective<br>I worry about how personal biological data might be used by authorities<br>I worry that personal biological data may not be treated confidentially<br>I worry about how my biological data and test results might be stored<br>I worry about how my personal biological data might be used by personalised nutrition providers<br>I worry about how my personal biological data might be used by insurance companies<br>I worry that my personal biological data could be accessed by hackers | Five-point scale, from completely disagree to completely agree | .924 |

**Table 2. Number of participants and words in each condition.**

| Sample condition | Urine | | Blood | | Stool | |
|---|---|---|---|---|---|---|
| Certainty condition | Certainty | Uncertainty | Certainty | Uncertainty | Certainty | Uncertainty |
| Participants n= | 62 | 32 | 63 | 18 | 58 | 28 |
| Total number of words produced | 185 | 95 | 188 | 53 | 173 | 83 |
| Mean number of words produced per participant | 2.98 (SD = 0.13) | 2.97 (SD = 0.18) | 2.98 (SD = 0.13) | 2.94 (SD = 0.24) | 2.98 (SD = 0.13) | 2.96 (SD = 0.19) |

**Table 3. Word categories, descriptors, and exemplars.**

| Category | Description of category. Word… | Exemplar words | n (% of total) |
|---|---|---|---|
| Disgust | Indicates repulsion, abhorrence | Disgusting, Gross, Yuk, Nasty | 57 (7.3) |
| Synonym | Is a synonym for the sample | Poo, Wee, Blood | 44 (5.7) |
| Straightforward | Refers to ease or simplicity of task | Simple, Easy, Straightforward, Manageable | 138 (17.8) |
| Convoluted | Suggests the task is onerous or lengthy | Inconvenient, Hassle, Involved, Tedious, Tricky | 81 (10.4) |
| Worry | Refers to anxiety or nervousness | Worry, Anxious, Nervous | 19 (2.4) |
| Smell | Refers to odour | Smell, Stink | 14 (1.8) |
| Hygiene | Refers to aspects of hygiene and sterility | Contamination, Sterile, Unhygienic, Cleanliness | 36 (4.6) |
| Dirt | Is a descriptor of dirt or mess | Messy, Wet, Sloppy, Dirty, Wee on hands | 69 (8.9) |
| Intrusive | Indicates procedure is intrusive or (too) personal | Embarrassing, Vulnerable, Private, Intrusive | 23 (3.0) |
| Negative experience | Refers to negative experience | Awkward, Uncomfortable, Weird, Unpleasant | 47 (6.0) |
| Rejection | Indicates unwillingness to carry out procedure | No way, No, Never, Too much | 28 (3.6) |
| Reference to text | Refers to the instructions in the sample condition | Spoon, Post, Spatula, Mid-stream, Lancet | 40 (5.1) |
| Worthwhile | Suggests process is useful or beneficial | Helpful, Necessary, Interesting, Good | 82 (10.6) |
| Reference to experience | Refers to previous experience/ familiarity with procedure | Diabetes, Covid, Bowel cancer test, Used to it | 6 (0.8) |
| Needles | Refers to needles | Needle, Prick | 5 (0.6) |
| Pain | Refers to pain | Pain, Ouch, Painful | 16 (2.1) |
| Science | Refers to science or medicine | Science, Lab, Scientific, Doctors | 29 (3.7) |
| Health | Refers to health or wellbeing | Health, Healthy | 10 (1.3) |
| Privacy | Refers to privacy or privacy concerns | Privacy, Confidentiality, DNA access, Biological ID | 9 (1.2) |
| Novel | Refers to newness or innovation | New, Innovative, Modern | 10 (1.3) |
| Other | Cannot be assigned to any other code | N/A, Cost, Trying, | 14 (1.8) |

**Categories of words produced under certainty.** Across all three sample conditions, the prevalence of word categories appeared to be related to the (un)certainty conditions. When the science was presented as certain, participants gave more positive responses to the idea of providing any kind of personal biological sample for the purposes of PN, whereas when the science was presented as less certain, rather less positively connoted responses were given. Categories with more positive connotations, such as 'straightforward' and 'worthwhile', featured more frequently in the certainty condition. Conversely, categories with more negative connotations, such as 'disgust' and 'convoluted', tended to feature more frequently in the uncertainty condition. Category frequencies for the two certainty conditions, where responses accounted for at least 5% of total responses in either condition are shown in Fig 1 below.

**Categories of words produced in the sample conditions.** Some categories (e.g., 'convoluted', and 'worthwhile') were present in relatively large numbers across all the sample conditions. Others were solely, or much more highly represented in only one condition (e.g., 'worry', 'pain', and 'needle' were almost exclusively

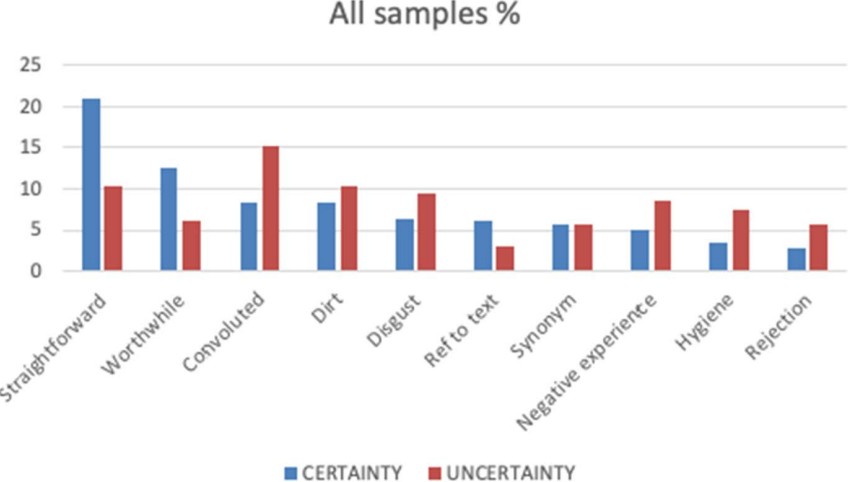

**Fig 1. Category frequency in Certainty and Uncertainty conditions – all sample types.** Expression of certainty also had an impact on the median affective response to the words provided. There was a statistically significant difference between the median affective responses in the two (un)certainty conditions. The median rank affect score for participants in the Certainty condition was 138.71 – more positive than neutral – and in the Uncertainty condition 88.36 – more negative than neutral ($\chi2(1) = 25.04$, $p < 0.010$).

associated with the Blood condition, and 'smell' with the stool condition). There were some clear differences in the frequencies of the produced word categories across the three conditions. For example, stool was considered much less 'straightforward' than urine and blood. This pattern was reversed for 'disgust' where stool featured much more than the other two sample types. Thus, whilst the idea of giving blood and urine samples provokes some negative responses, they are overall considered more favourably than giving a stool sample. This is illustrated in Fig 2 below. Categories are included where responses accounted for at least 5% of total responses in any of the three sample conditions.

## Study 2: Results

Table 4 details the main effects of the (un)certainty manipulation for each of the dependent variables that met the threshold of $p < .05$. All except two of the ethical variables – equality of access to PN and data privacy and security – were statistically significant with a similar pattern of results: veracity of PN claims, attitudes toward PN, perceived benefit, intention to take up PN, and trust in efficacy and regulation were all seen more positively by those in the certain condition than those in the uncertain condition. Similarly, participants in the certainty condition indicated they were less concerned about ethical issues and held more positive views towards PN than participants in the uncertainty condition. There were no significant main effects of the sample condition.

In summary then, in response to the relatively subtle manipulations of uncertainty within the vignette, both the words produced and the associated affect ratings in the thought association task and the responses to the survey questions, indicated that greater certainty was associated with greater positivity about PN. The findings from the word association study suggests different samples led to the production of different word categories and that the affect attached to these words was more negative for stool samples than it was for blood and urine samples. There was a different pattern of results for the survey questions relation to the sample manipulation. There were no significant differences between the responses to questions about blood, urine or stool samples on any of the 7 PN survey measures.

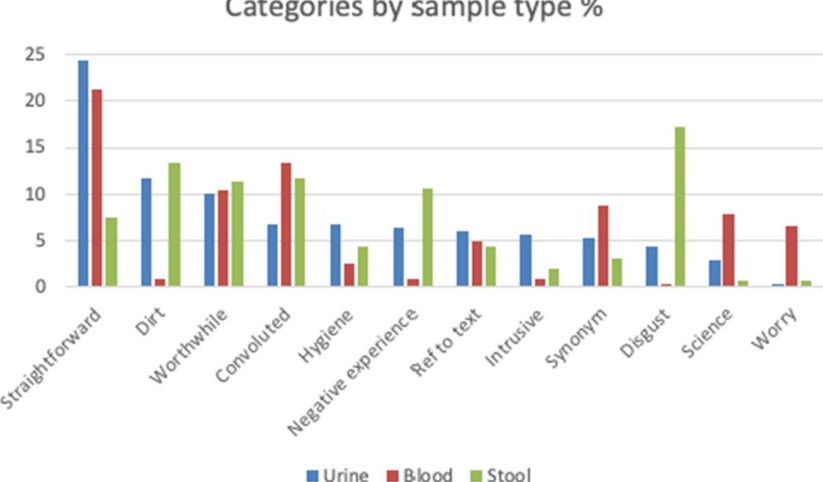

**Fig 2. Category frequencies by sample condition.** A Kruskal-Wallis H test showed that there was a statistically significant difference between the medians of the affect ratings for the three types of samples, $\chi2(2) = 12.38$, $p = 0.002$. Further Bonferroni-corrected pairwise comparisons show a statistically significant difference between stool and urine samples ($\chi2(2) = 29.51$, $p = 0.021$) and stool and blood samples ($\chi2(2) = 11.39$, $p = 0.003$) but not between urine and blood samples. The collection of stool samples evoked a more negative affect rating (median rank of 101.75) than blood (139.65) or urine (131.26).

**Table 4. Significant Kruskal-Wallis effects of Certainty (at p<.05 level), df=1.**

| Dependent variable | Condition | Median Rank | N | H | p value | 95% Confidence Interval | |
|---|---|---|---|---|---|---|---|
| | | | | | | Lower Bound | Upper Bound |
| **Ethical variables** | | | | | | | |
| Veracity of PN claims | Certainty | 143.37 | 181 | 21.26 | <0.001 | 2.41 | 2.64 |
| | Uncertainty | 96.9 | 77 | | | 1.88 | 2.27 |
| **PN variables** | | | | | | | |
| Attitude toward PN | Certainty | 154.99 | 180 | 78.11 | <0.001 | 4.10 | 4.26 |
| | Uncertainty | 65.77 | 76 | | | 3.15 | 3.46 |
| Perceived benefit of PN | Certainty | 150.76 | 182 | 50.65 | <0.001 | 4.20 | 4.36 |
| | Uncertainty | 80.94 | 77 | | | 3.30 | 3.68 |
| Intention to take up PN | Certainty | 152.5 | 180 | 57.53 | <0.001 | 3.36 | 3.59 |
| | Uncertainty | 76.42 | 78 | | | 2.24 | 2.65 |
| Trust/efficacy of regulation | Certainty | 117.78 | 179 | 9.73 | 0.002 | 2.79 | 3.09 |
| | Uncertainty | 149.3 | 74 | | | 2.27 | 2.76 |

## Discussion

The studies presented here explored the impact of certainty and sample type on affective responses and attitudes to PN. Communication of uncertainty, even though implied rather than explicit, influenced participant responses in a word association task and in survey measures of attitudes to personalised nutrition. Across both studies, where the science was presented as certain, positive word categories were more frequent, the associated affect more positive, and attitudes to PN more favourable. Consideration of sample type affected affective responses, but not attitudes towards PN.

The impact of communicating uncertainty about PN has, to the best of our knowledge, thus far been unaddressed. This is important as, even though commercial presentations of the benefits of PN may tend towards conveying certainty [45], the scientific evidence base underlying nutrition recommendations is at a relatively early stage and PN recommendations based on biological samples does necessarily yield superior individual outcomes [46]. Our aim was to explore the impacts of uncertainty on both qualitative (study 1) and quantitative (study 2) responses. The vignettes to which participants were responding were framed to either emphasise certain knowledge or a gap in knowledge about PN and its impacts. This is characterised by Gustafson and Rice [27] as 'deficient uncertainty' – an uncertainty type previously associated with a mix of positive, negative, and null main effects. In this case, the findings were clear – when knowledge about personalised nutrition was framed as less certain, perceptions of providing stool samples were more negative. Conversely, more certain knowledge about PN was followed by less negative assessments of stool samples. The results of the questionnaire study also indicated the impact of the (un)certainty frame. Information about personalised nutrition presented as more certain was related to more positive attitudes to PN, to its perceived benefit and efficacy, and associated with greater intention to engage with PN and greater trust in regulation. The clear pattern of the impacts of uncertainty in both qualitative and quantitative outcomes are particularly noteworthy bearing in mind the subtle nature of the manipulation embedded in the vignette that participants were responding to. Being certain or uncertain was not mentioned at all, rather it was a case of using more definitive or provisional language in the claims made about PN. Extending the reasoning outlined in [47] this suggests that the uncertainty frame increases the 'hypothetical distance' of PN, rendering it less tangible and relevant and consequently perceived less positively.

The effects of the nature of the sample were more mixed. In the word association task and the associated affect ratings there were clear differences in the reactions to samples – most notably stool samples were considered more negatively than samples of urine or blood. This was most obviously manifest in high levels of disgust and low ratings of the straightforwardness of the procedure. However, there was no effect of sample type on attitudes to personalised nutrition.

Previous research has found negative responses to stool samples, but these have largely been in the context of testing where there are potentially deleterious personal implications, such as for colorectal cancer [48], for biobanks where sample donation is more likely to be motivated by reciprocity or altruism [49] or assessments of community prevalence of antibiotic resistant organisms where there is little or no benefit to the individual [50]. In the context of PN, the results were mixed. In the thought listing task, many of the categories in which words were generated were sensitive and specific to the different sample types. Stool samples were notably different from blood and urine samples having the relatively high and low percentage of responses around 'disgust' and 'straightforward' respectively. Similarly, the overall affect associated with the words in each category was significantly more negative for stool samples than for either blood or urine. However, in contrast to both the thought listing and the associated affect ratings, responses to the outcome measures in the questionnaire were no more negative in the stool sample condition than in the blood or urine sample conditions. Perhaps the most likely explanation for this is that for a general population sample, the negativity that followed being asked to consider stool sampling was primarily an affect-based response which was successfully captured in the thought listing tasks. In contrast, in the questionnaire study, the focus moved from a detailed consideration of affect associated with a particular sample to its association with a range of assessments relating to PN. It is these considerations that are foregrounded in the questionnaire with the consideration of the sample procedure itself relegated to the background.

## Strengths and limitations

Although the first study highlighted the impacts of inferring uncertainty on perceptions of personalised nutrition, other studies have noted the importance of both the source of the information [51] and/or the characteristics of the respondent [33]. We did not consider this and future studies exploring the impact of uncertainty on perceptions of PN could usefully incorporate these established moderators and mediators. Nor did we consider the impact of prior experience with or knowledge about PN, or of demographic differences, on responses to PN. These could be addressed in future studies exploring both

perceptions of PN and of claims made about its efficacy. In retrospect we might have usefully incorporated a more explicit theoretical consideration of disgust given that faeces are a 'universal disgust object' [52] and research has highlighted the role of disgust reactions in reducing willingness to provide a stool sample in the context of testing for serious illness such as colorectal cancer [53,54].

## Implications for communication and further research

Subtle cues framing the value of PN as more certain clearly led to greater positivity about PN itself, its regulation, and the likelihood of personal engagement as well as less negative affect around providing samples. This is challenging given that there is still considerable uncertainty around the science itself and the exact nature of the links between the analysis of biospecimens and the impact of consequent dietary recommendations. An analysis of the provision of gut microbiome-based services on the internet explored how companies articulated these uncertainties [45] in the information provided on their websites. The general mode of uncertainty communication was to make unequivocal claims about the gut microbiome and its links to disease and the science underlying their testing regimens, and then – often in the small print – to issue specific disclaimers that the service offered was not medical or that some analytic errors were unavoidable. We have seen overall that clear, though subtle, communication of uncertainty reduces consumer perception of the benefits of PN. Further work might usefully use eye-tracking and think aloud methods focusing on websites promoting analysis of the microbiome to promote PN products, to identify how consumers understand unequivocal claims of benefit juxtaposed with disclaimers, whether and how this conveys uncertainty, and characterising the impacts of that uncertainty on likely product purchase.

Providing a biological sample is necessary for precise personalised nutrition, so for those that aim to increase the uptake of PN, understanding how individuals perceive the donation of biospecimens can help identify barriers and by reducing them, ultimately increasing adoption. The results of both studies might lead to the conclusion that in the context of seeking to promote the benefits of microbiome based PN, there is value in de-coupling the stool sample collection/provision procedure from promoting appreciation of an (accurate) description of the benefits. A detailed description of the stool sampling procedure is necessary, given the primacy of hygiene considerations and the need for an uncontaminated sample; precise information about procedure and packaging and storage of the sample is required. Increasing willingness to provide a stool sample has been addressed by others – with varying success – in contexts where the potential benefits of doing so are more certain. These include minimizing repetition of words like faeces in invitation letters for colorectal cancer screening [54] and providing financial incentives [55]. Chen, Carpenter [56] took a highly participative approach with adolescents and young adults to developing effective and engaged sample provision strategies in the context of a longitudinal research study on the gut microbiome. The strategies themselves – for example – providing clear instructions and follow up communications and minimizing disgust and embarrassment are rather generic. However, their main point is the value of the highly participative approach taken to developing strategies and to maximising engagement in the research. These are likely to be considered less feasible in the commercial settings common in MBPN. Building on waste-water epidemiology, advanced technical solutions that eliminate the need for any hands-on engagement with faeces are also in development, for example, by separating the stool from wastewater after the toilet is flushed [57].

In conclusion, our findings have implications for PN in general and in particular for PN based on the microbiome. First, if consumer acceptance of PN based on provision of a biomedical sample is necessary for increased uptake [58], there exists a dilemma for those seeking to honestly communicate its benefits given the attendant – likely intractable – uncertainties. As Lohse [59] asks: "How can health care professionals frame uncertainty as a somewhat unavoidable part of precision medicine?". This is particularly challenging when many PN offerings are mediated through web-based commercial enterprises [60] where trust is often low and where interpersonal strategies for enhancing trust [61] and opportunities for supporting shared decision making [59] are limited. Set alongside this are the practical issues relating to ensuring that reluctance to provide a stool sample does not constrain those that wish to engage with microbiome based PN.

Further work to refine the impact of communicative and product design strategies that minimize barriers to engagement is required.

## Supporting information

**S1 Table. Demographic Information.**
(PDF)

## Acknowledgments

We appreciate the engagement of the participants that took part in the surveys for this study.

## Author contributions

**Conceptualization:** Katharine Lee, Julie Barnett.

**Formal analysis:** Katharine Lee, Estelle Corbett.

**Funding acquisition:** Julie Barnett.

**Investigation:** Katharine Lee.

**Methodology:** Katharine Lee, Rebecca Hafner, Julie Barnett.

**Supervision:** Julie Barnett.

**Writing – original draft:** Katharine Lee, Estelle Corbett, Julie Barnett.

**Writing – review & editing:** Katharine Lee, Estelle Corbett, Rebecca Hafner, Julie Barnett.

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
