## [Decision Letter · Decision Letter 0]

29 Jul 2025

Dear Dr. Lee,

Thank you for submitting your manuscript to PLOS ONE. After careful consideration, we feel that it has merit but does not fully meet PLOS ONE’s publication criteria as it currently stands. Therefore, we invite you to submit a revised version of the manuscript that addresses the points raised during the review process.

We look forward to receiving your revised manuscript.

Kind regards,

Tahir Turk, PhD

Academic Editor

PLOS ONE

Journal Requirements:

2 . Please provide additional details regarding participant consent. In the ethics statement in the Methods and online submission information, please ensure that you have specified what type you obtained (for instance, written or verbal, and if verbal, how it was documented and witnessed). If your study included minors, state whether you obtained consent from parents or guardians. If the need for consent was waived by the ethics committee, please include this information

“This project has received funding from the European Union’s Horizon 2020 research and innovation programme under grant agreement No 816303”

“NO authors have competing interests”

6. PLOS requires an ORCID iD for the corresponding author in Editorial Manager on papers submitted after December 6th, 2016. Please ensure that you have an ORCID iD and that it is validated in Editorial Manager. To do this, go to ‘Update my Information’ (in the upper left-hand corner of the main menu), and click on the Fetch/Validate link next to the ORCID field. This will take you to the ORCID site and allow you to create a new iD or authenticate a pre-existing iD in Editorial Manager.

Additional Editor Comments:

Unfortunately, we have only been able to identify one reviewer for your manuscript. However, we note very detailed feedback by the reviewer on ways to improve the manuscript for a second review noting that this is currently an under-studied topic for precision nutrition research. Please refer to the reviewer feedback below.

Reviewers' comments:

Reviewer's Responses to Questions

**Comments to the Author**

1. Is the manuscript technically sound, and do the data support the conclusions?

Reviewer #1: Yes

2. Has the statistical analysis been performed appropriately and rigorously?

Reviewer #1: Yes

3. Have the authors made all data underlying the findings in their manuscript fully available?

Reviewer #1: No

4. Is the manuscript presented in an intelligible fashion and written in standard English?

Reviewer #1: Yes

Reviewer #1: Summary: This paper sought to conduct a quasi-experimental study to understand the impact of certainty and sample type on responses and attitudes to personalized nutrition.

Reviewer’s Comments:

Overall:

1. This is an excellent paper investigating an under-studied topic part of precision nutrition research, that is important to understand in order to achieve success with PN based interventions.

Introduction

2. Page 3, Line 80-87: The authors discuss multiple definitions of PN, but do not formally define PN for their study. Perhaps this is by design but it would be good to explicitly state that the authors are not going to define PN for use in this study, and rather the study aims to assess the pre-existing conceptualization of “personalized nutrition” at this point, without defining it. (See comment 4).

3. Page 5, Line 140-155: It is true that PN is a relatively nascent field (as noted in the Discussion) and there is a lot of uncertainty, but several human studies exist that the authors could cite as adding to the complexity. For example, Zeevi et al. 2015 (Cell) is motivating the NIH-funded Nutrition for Precision Health study in the United States, which is currently ongoing and aims to enroll 8000+ people to test responses to diet. Popp et al. 2022 (JAMA) tried to apply the same algorithm used in Zeevi et al. 2015 (Cell) but for weight loss rather than glucose control, and found null results unlike Zeevi which is an interesting discussion point. The PREDICT-1 and other PREDICT studies were also done in thousands of humans. Discussion of these and other studies, even if to highlight the uncertainties these studies may raise in terms of human work, should be considered for this paragraph.

Methodology & Results

4. Page 8, Lines 238: How were participants recruited – targeted emails, a list serv, posters?

5. Page 8, Lines 238: How do you know participants are nationally representative?

6. Page 8, Line 248: What is a median age of 45-54? The median should be one number if representing the 261 individuals. Is this the IQR? Or is this 2 medians for the 2 different studies?

7. Either in main text or supplementary material, it would be helpful to have a table of characteristics for the participants of this study (unless age was the only demographic characteristic collected).

8. Page 8, Lines 238-248: Did you assess the level of participants’ baseline experience or knowledge with PN prior to the study? Such as, have they used mobile apps purporting to be based on precision nutrition? Was personalized nutrition defined for them in some way? I’m curious if participants understood what was meant by personalized nutrition – as simply one’s nutritional needs for height, weight, sex, physical activity – or with the additional variables mentioned in the introduction section that get closer to true individualization? Additionally, education level may influence results – see comment 8 below. I might hypothesize that those who say PN is ‘certain’ may not be as well educated about PN or nutrition in general?

9. Page 8, Lines 247-248: Participant median age of 45-54 – it would be interesting to break down the results by age group and see if there are differences between younger and older groups, considering openness to new technologies or concepts varying with age (I say this anecdotally)

10. Page 12: A Kruskal Wallis test is meant to test for differences between medians, not means. Elsewhere “median” is noted, though, so please check this is consistent throughout.

11. It would be useful to perhaps in Supplementary material, include the NHS guidelines for collecting the samples especially since the authors adapted them for this study. Adapted in what way? Identifying or accessing the specific guidelines used might not be straightforward for those outside the UK.

Discussion:

12. Regarding comment 7, I see a lack of background info is noted in the Strengths and Limitations – it might be good to also note the types of characteristics that would be relevant to measure, and how they might be impact the results, even though you did not measure this.

References:

13. Again please cite the major human studies in the field including Zeevi et al, etc. (see above).

[While revising your submission, please upload your figure files to the Preflight Analysis and Conversion Engine (PACE) digital diagnostic tool, https://pacev2.apexcovantage.com/ . PACE helps ensure that figures meet PLOS requirements. To use PACE, you must first register as a user. Registration is free. Then, login and navigate to the UPLOAD tab, where you will find detailed instructions on how to use the tool. If you encounter any issues or have any questions when using PACE, please email PLOS at figures@plos.org

---

## [Author Response · Author response to Decision Letter 1]

9 Oct 2025

Please see attached response letter and table.

---

## [Decision Letter · Decision Letter 1]

15 Oct 2025

Public perceptions of biospecimen sampling and uncertainty in the context of personalised nutrition

PONE-D-25-20637R1

Dear Dr. Lee,

We’re pleased to inform you that your manuscript has been judged scientifically suitable for publication and will be formally accepted for publication once it meets all outstanding technical requirements.

Kind regards,

Tahir Turk, PhD

Academic Editor

PLOS ONE

Additional Editor Comments (optional):

Reviewers' comments:

Reviewer's Responses to Questions

**Comments to the Author**

Reviewer #1: All comments have been addressed

2. Is the manuscript technically sound, and do the data support the conclusions?

Reviewer #1: Yes

3. Has the statistical analysis been performed appropriately and rigorously?

Reviewer #1: Yes

4. Have the authors made all data underlying the findings in their manuscript fully available?

Reviewer #1: Yes

5. Is the manuscript presented in an intelligible fashion and written in standard English?

Reviewer #1: Yes

Reviewer #1: Thank you; All comments are addressed. I have no other comments.

 ............................................

**Do you want your identity to be public for this peer review?** For information about this choice, including consent withdrawal, please see our Privacy Policy

Reviewer #1: No

---

## [Editor Report · Acceptance letter]

PONE-D-25-20637R1

PLOS ONE

Dear Dr. Lee,

I'm pleased to inform you that your manuscript has been deemed suitable for publication in PLOS ONE. Congratulations! Your manuscript is now being handed over to our production team.

Kind regards,

on behalf of

Dr. Tahir Turk

Academic Editor

PLOS ONE